# Barriers to Healthcare Access During the Coronavirus Disease 2019 (COVID-19) Pandemic: A Cross-Sectional Study Among Romanian Patients with Chronic Illnesses and Confirmed SARS-CoV-2 Infection

**DOI:** 10.3390/healthcare13111333

**Published:** 2025-06-03

**Authors:** Adrian Militaru, Petru Armean, Nicolae Ghita, Despina Paula Andrei

**Affiliations:** 1“St. John” Clinical Hospital, Faculty of General Medicine; Clinical Department of Urology, “Carol Davila” University of Medicine and Pharmacy, 13 Vitan-Bârzești Road, Sector 4, 042122 Bucharest, Romania; adrian.militaru@drd.umfcd.ro; 2“Prof. Dr. Theodor Burghele” Clinical Hospital, Faculty of Midwifery and Nursing, Department of Social Medicine, Clinical Department of Internal Medicine, “Carol Davila” University of Medicine and Pharmacy, 20 Panduri Road, 061344 Bucharest, Romania; petru.armean@umfcd.ro; 3“Prof. Dr. Theodor Burghele” Clinical Hospital, Faculty of Midwifery and Nursing, Clinical Department of Internal Medicine, “Carol Davila” University of Medicine and Pharmacy, 20 Panduri Road, 061344 Bucharest, Romania; nicolae.ghita@drd.umfcd.ro; 4“Elias” University Emergency Hospital, Faculty of Midwifery and Nursing; Clinical Department of Neurosurgery, “Carol Davila” University of Medicine and Pharmacy, 17 Mărăști Boulevard, 011461 Bucharest, Romania

**Keywords:** COVID-19, chronic diseases, healthcare access, patient satisfaction, digital divide, remote consultations, rural health, Romania, healthcare barriers, public health resilience

## Abstract

Background/Objectives: The COVID-19 pandemic presented unprecedented challenges to healthcare systems worldwide, significantly impacting individuals with chronic conditions who depend on continuous medical care. In Romania, the pandemic revealed systemic vulnerabilities, particularly in ensuring access to services for older adults and rural populations. This study aimed to assess perceived barriers to healthcare access and service quality among Romanian patients with chronic diseases and a confirmed history of COVID-19, within the framework of the country’s multi-tiered healthcare system. Methods: A cross-sectional study was conducted between January and March 2025, involving 16 adult participants diagnosed with at least one chronic illness. Data were collected using a 30-item questionnaire administered by the principal investigator after obtaining informed consent. The instrument explored access to services, challenges related to remote consultations, and satisfaction with nursing care. Descriptive and comparative analyses were carried out based on age group and area of residence. Due to the small sample size, the results are considered exploratory and context-specific. Results: Most participants reported disrupted access to healthcare services, especially within public sector facilities. Rural residents experienced longer delays in receiving care than those in urban areas. Digital health tools were perceived as barriers by 75% of respondents aged 60 and above, while younger participants adapted more easily. Overall satisfaction with nursing care was moderate to high (mean score: 3.56/5), with the highest ratings observed among patients aged 30–60 years. Conclusions: This study highlights significant barriers to healthcare access among Romanian patients with chronic illnesses and a confirmed COVID-19 diagnosis during the pandemic. The key challenges included digital exclusion and rural–urban disparities. The findings underscore the need for targeted strategies to enhance digital health literacy, adapt care delivery models, and strengthen healthcare system resilience in future public health emergencies.

## 1. Introduction

Chronic diseases remain a major public health challenge worldwide, including in Romania. Their management typically spans all three tiers of healthcare: primary, secondary, and tertiary care. In Romania, primary care physicians oversee the continuous monitoring and treatment of most chronic conditions, prescribing maintenance therapies and tracking patient evolution [1,2]. However, the confirmation of diagnosis and initiation of specialized treatments often require consultations in outpatient secondary care settings, while complex cases necessitate hospitalization in tertiary care centers [1].

The COVID-19 pandemic severely disrupted healthcare systems globally, with particularly adverse effects on individuals managing chronic illnesses [3]. In Romania and other countries, routine medical services—including consultations, diagnostic evaluations, and follow-up care—were postponed, redirected, or suspended to prioritize COVID-related responses [4,5]. These interruptions significantly burdened individuals with long-term conditions who depend on uninterrupted access to care for disease control and survival [6,7].

Multiple barriers to healthcare continuity emerged during the pandemic, including medical staff shortages, system reorganization, and fear of exposure among patients [8]. These disruptions disproportionately affected vulnerable populations—especially the elderly, rural residents, and those with low digital literacy—further exacerbating pre-existing inequalities in healthcare access [9,10].

To address these challenges, healthcare systems increasingly relied on digital tools such as telemedicine, video consultations, and e-prescriptions [11]. While such tools provided alternatives to face-to-face care, they also introduced significant accessibility challenges for older adults and individuals from disadvantaged backgrounds lacking the necessary devices, skills, or internet connectivity [12,13]. This “digital divide” has become a defining factor in healthcare equity during health crises [14].

In Romania, little is known about how patients with chronic illnesses navigated the healthcare system during the pandemic. The existing literature has focused primarily on the clinical aspects of COVID-19, leaving gaps in knowledge about service accessibility, patient experiences, and system-level barriers for non-COVID conditions [15,16,17]. Capturing the lived experiences of these patients is critical for developing inclusive, resilient healthcare policies.

Moreover, chronic diseases such as cancer, diabetes, and cardiovascular conditions are already associated with high mortality risk. When compounded by COVID-19, survival rates decline even further. Consequently, few individuals met the dual inclusion criteria of chronic illness and confirmed SARS-CoV-2 infection and survived, limiting the size of the eligible study population [18]. Data were collected in early 2025, after the acute pandemic phase had passed, over a short timeframe—further justifying the exploratory nature of this research.

Small-sample exploratory studies have been used in similar post-crisis contexts to investigate healthcare access barriers and patient experiences [19,20,21]. While they may lack statistical power, such studies can offer early, context-specific insights that help guide policy and preparedness planning under exceptional circumstances.

This study aims to evaluate the accessibility and perceived quality of healthcare services among Romanian adults living with chronic conditions and a confirmed history of COVID-19, with a particular focus on digital access, rural–urban disparities, and care continuity challenges.

## 2. Materials and Methods

### 2.1. Study Design and Setting

This cross-sectional study aimed to evaluate healthcare accessibility and perceived service quality among Romanian adults living with chronic illnesses and a confirmed history of COVID-19 during the pandemic. The principal investigator, based in Bucharest, conducted the data collection between January and March 2025. Depending on public health conditions and participant availability, questionnaires (Appendix A) were administered either in person or by telephone, in accordance with infection prevention protocols. All interviews were audio-recorded after obtaining informed consent to ensure the accuracy of responses.

### 2.2. Participants and Inclusion Criteria

Participants were recruited through convenience and snowball sampling, primarily from patients attending specialty outpatient consultations. Only a few were post-hospitalization survivors, as mortality was high among individuals with chronic illnesses co-infected with SARS-CoV-2 during the acute phase of the pandemic. Data collection took place in the early months of 2025, after the acute phase had passed, within a limited timeframe. In this post-crisis context, identifying patients who simultaneously met all the eligibility criteria proved challenging due to high mortality rates among comorbid patients, limited healthcare access, logistical constraints, and socioeconomic barriers.

The inclusion criteria were as follows: (1) age 18 or older; (2) confirmed diagnosis of at least one chronic condition (hypertension, ischemic heart disease, diabetes mellitus, cancer, or chronic obstructive pulmonary disease); and (3) confirmed history of SARS-CoV-2 infection prior to enrollment. A total of 16 individuals were enrolled, with ages ranging from 30 to over 80 years. The sample was diverse in terms of age, gender, area of residence, and socioeconomic background.

Given the strict inclusion criteria and the post-crisis timing of data collection, the final sample size was limited. This small yet diverse cohort reflects the real-world difficulty of identifying and recruiting individuals who had survived both chronic disease and COVID-19 in the Romanian healthcare system. Similar exploratory studies with reduced sample sizes have been conducted in post-pandemic or resource-limited settings to generate early, context-specific evidence [19,20,21].

### 2.3. Data Collection Instrument

Data were collected using a 30-item questionnaire developed by the research team. The questionnaire was administered by the principal investigator after obtaining written or verbal informed consent from each participant. It covered the following domains:Accessibility of medical services during the pandemic.Delays in medical care and their perceived impact on health.Use of public versus private healthcare providers.Experience with remote consultations.Satisfaction with nursing care.

Although initially designed to include both open- and closed-ended items, the final version consisted primarily of closed-ended questions. All the interviews were conducted in Romanian, and audio recordings were made to ensure the accuracy of responses. After transcription and anonymization, the data were entered into a Microsoft Excel database for analysis.

### 2.4. Statistical Analysis

Quantitative data were analyzed using descriptive and comparative statistics. Categorical variables are reported as absolute frequencies and percentages, while continuous variables are presented as means and standard deviations. Subgroup comparisons were performed based on age group (<60 vs. ≥60 years) and area of residence (urban vs. rural). Statistical significance was defined as *p* < 0.05. All the analyses were conducted using Microsoft Excel and standard statistical functions.

### 2.5. Ethical Considerations

This study was conducted in accordance with the Declaration of Helsinki [22] and relevant Romanian legislation, including Law No. 190/2018 and the EU General Data Protection Regulation (GDPR) 2016/679. All the participants received clear and detailed information about this study’s purpose and procedures and provided both written and verbal informed consent before participation. The study protocol was approved by the Ethics Committee of Prof. Dr. Theodor Burghele Clinical Hospital (Approval No. 4/12.10.2020). Participant confidentiality and anonymity were strictly protected, and no personal identifiers were collected, except for optional email addresses, which were securely encoded.

### 2.6. Use of Artificial Intelligence Tools

No generative artificial intelligence (GenAI) tools were used to generate or analyze any data in this study. Minor language editing assistance was provided using AI-based tools; however, all the final content was manually reviewed and edited by the authors, who take full responsibility for the accuracy and integrity of this manuscript.

## 3. Results

A total of 16 participants met the eligibility criteria and completed this study. The majority were female (62.5%), and most resided in urban areas (68.7%). The age range spanned from 30 to over 70 years. These sociodemographic details are presented in Table 1.

The participants were asked which healthcare sector was more accessible during the COVID-19 pandemic (Question 14). Half of the respondents (8 out of 16) primarily accessed public healthcare services, while the other half relied on private providers. The shift toward private care was often attributed to service unavailability or long delays in the public sector (Table 2).

Regarding digitalization (Question 23), 68.7% of the participants (11/16) stated that digitalization had no impact on their access to services, while 31.3% (5/16) reported improved access. Notably, no participants indicated that digitalization had a negative effect (Table 3), which contrasts with the expectations of a pronounced digital divide.

When asked whether delays in medical consultations and procedures negatively affected their health (Question 21), 7 out of 16 participants (43.8%) responded affirmatively. They described worsening symptoms, missed follow-ups, and emotional distress.

The perceived quality of nursing care during the pandemic was evaluated using a 5-point Likert scale (Question 30). The average rating was 3.25, indicating moderate satisfaction. The most frequent scores were 4 (six respondents), 3 (five respondents), and 2 (four respondents), while only one respondent gave the highest rating of 5 (Table 4).

These findings reflect diverse healthcare experiences across the sample and provide insights into the accessibility challenges encountered by Romanian patients living with chronic illnesses and a confirmed history of SARS-CoV-2 infection. The numeric data support a broader understanding of how public vs. private service structures and digital tools interacted with patient needs during the pandemic.

## 4. Discussion

This study offers timely, context-specific insights into the challenges faced by Romanian patients with chronic conditions and a confirmed SARS-CoV-2 infection in accessing healthcare during the COVID-19 pandemic. The main barriers identified were delays in consultations, limited availability of public healthcare services, and restricted digital access—particularly among older adults and rural residents. These findings are consistent with previous European research, which indicates that the pandemic exacerbated existing health inequities, disproportionately impacting vulnerable populations [3,20,21].

Half of the participants (8 out of 16) reported using private healthcare services during the pandemic, while the other half relied on the public sector. For many, the shift toward private care was prompted by delays or service unavailability in public institutions. These results mirror trends observed in other European healthcare systems, where public sector strain led to greater out-of-pocket spending and widened socioeconomic disparities in access [4,23].

Although remote consultations were introduced as an emergency solution to ensure continuity of care, access to these services remained uneven. Among participants aged 60 years or older, 75% viewed digital health tools as barriers to care, compared with only 25% of younger respondents (Question 23). This reflects a persistent digital divide, well-documented in the international literature, particularly among older adults or individuals with limited digital literacy and Internet access [11,23,24].

The overall satisfaction with nursing care was moderate to high (mean = 3.56/5), with the highest ratings reported by participants aged 30–60 years (mean = 4.1). Older participants (>60 years) expressed lower satisfaction (mean = 3.2), potentially due to reduced in-person interaction, limited digital proficiency, or staff shortages. This trend is consistent with recent hospital-based studies conducted during the pandemic [13,25].

Notably, 7 out of 16 participants (43.7%) reported that delays in consultations and procedures negatively affected their health (Question 21), including worsening symptoms and delayed diagnoses. However, 12 respondents (75%) stated that their health improved after receiving care, while the remaining 25% reported no change (Question 29). No participants reported a decline in health status, suggesting that, despite perceived negative effects of delays, clinical care—once accessed—remained effective.

These findings highlight the urgent need to address both structural and digital barriers in healthcare systems, especially regarding chronic disease management during public health emergencies. Strengthening infrastructure, enhancing service responsiveness, and ensuring equitable access to digital care are critical elements for improving resilience in future crises.

However, these conclusions should be interpreted with caution due to the small sample size and the exploratory nature of this study, as further discussed in the Section 4.1.2.

### 4.1. Strengths and Limitations

#### 4.1.1. Strengths

This study offers timely and context-relevant insights into the challenges faced by Romanian patients living with chronic diseases and a confirmed SARS-CoV-2 infection during the COVID-19 pandemic. By addressing both digital and geographic disparities, the analysis provides a nuanced understanding of structural barriers to healthcare access. The inclusion of participants with laboratory-confirmed COVID-19 adds specificity and increases this study’s relevance for future public health emergency preparedness. Additionally, the integration of quantitative results with direct patient feedback—such as perceptions of care delays and nursing satisfaction—supports a patient-centered interpretation of the findings.

#### 4.1.2. Limitations

The primary limitation of this study is the small sample size (*n* = 16), which limits statistical power and restricts the generalizability of the findings. However, this reflects the real-world challenges in identifying eligible participants after a major public health crisis. Chronic conditions such as cancer, diabetes, and cardiovascular disease are already associated with high mortality, and when combined with a systemic infection like COVID-19, survival rates decline even further. Data collection occurred in early 2025, following the acute phase of the pandemic, during a constrained timeframe in which only a small number of survivors meeting the strict inclusion criteria could be identified—particularly among older adults and individuals with multiple comorbidities.

Despite these constraints, small-sample exploratory studies have been widely employed in post-crisis and resource-limited contexts to investigate healthcare access and patient experiences [19,20,21]. While limited in scale, such studies can offer meaningful early-stage insights to inform preparedness and policy development in exceptional circumstances.

Another limitation concerns the use of convenience and snowball sampling methods, which may have introduced selection bias—potentially over-representing individuals with higher digital literacy or greater engagement with the healthcare system. Most participants were recruited through specialized outpatient consultations, as post-hospitalization survivors were fewer and more difficult to reach.

Although the data were not self-reported in the traditional sense, they were collected directly by the principal investigator through face-to-face or telephone-administered questionnaires, minimizing the risk of misinterpretation and enhancing data accuracy. Demographic questions were positioned at the beginning of the questionnaire to reduce participant discomfort and prevent concern about geographic traceability. County-level data were intentionally omitted to safeguard participant anonymity, particularly for individuals from rural or underserved areas.

Finally, the cross-sectional design limits the ability to establish causal relationships. Moreover, the absence of clinically validated outcome measures restricts the capacity to objectively assess the health impact of delayed care.

### 4.2. Policy Implications

The findings of this study underscore the urgent need for targeted health policy measures to promote equity and continuity of care for individuals with chronic conditions during public health emergencies. The COVID-19 pandemic revealed significant disparities in healthcare access, particularly affecting older adults, rural populations, and those with limited digital literacy.

One key recommendation is to develop hybrid care models that integrate in-person consultations with remote service options, ensuring that digital health tools complement rather than replace direct interactions between patients and providers. In the Romanian context, such digital services must be supported by user-friendly platforms, caregiver involvement for older patients, and simplified procedures for electronic documentation and prescriptions [26].

Bridging the digital divide identified in this study also requires investment in targeted digital literacy initiatives. These programs should prioritize training for older adults, informal caregivers, and socioeconomically vulnerable groups to ensure they can effectively access and navigate digital health platforms [12,27].

To strengthen healthcare system resilience, public health authorities should establish contingency protocols that protect access to chronic disease services during emergencies. This includes safeguarding outpatient consultation slots for non-COVID patients, maintaining continuity in primary care, and enabling remote prescription renewals.

Finally, investing in public health infrastructure—particularly in underserved rural regions—is essential to reducing territorial disparities and restoring trust in the public healthcare system. Strategies such as expanding broadband Internet coverage, deploying mobile health units, and supporting community-based care programs may serve as sustainable solutions for enhancing both access and system preparedness [23].

## 5. Conclusions

This exploratory study highlights several critical barriers to healthcare access among Romanian patients with chronic conditions and a confirmed history of COVID-19. Delays in consultations and medical procedures—particularly in the public sector—were frequently reported and perceived as having a negative impact on health outcomes. Geographic and demographic disparities were also evident, with rural residents and older adults facing greater challenges in accessing timely and appropriate care.

Although digital health tools provided alternative pathways for service delivery, they also created significant accessibility issues, especially for individuals with low digital literacy or limited access to technology. These findings emphasize the importance of designing inclusive digital health strategies that actively support vulnerable populations, particularly older adults and those in rural areas.

The overall satisfaction with healthcare services was moderate to high, with younger adults reporting greater satisfaction than their older counterparts. This variation suggests opportunities to improve patient–provider communication and adapt service delivery to better meet the needs of different demographic groups.

Given the small sample size and the exploratory nature of this study, the conclusions presented here should be interpreted with caution. However, comparable small-sample studies have been successfully used to generate early, context-specific evidence in post-crisis settings [19,20,21]. Despite its limitations, this study offers valuable insights into healthcare system vulnerabilities and patient experiences that can inform future health policy development.

To strengthen healthcare equity and resilience in future public health emergencies, targeted interventions should aim to achieve the following:Reduce rural–urban disparities in access to care;Improve digital health literacy among underserved groups;Ensure continuity of care for patients with chronic illnesses;Develop hybrid care models that integrate digital and in-person services effectively.

Ensuring equitable access to healthcare and maintaining service quality must remain central priorities in both emergency preparedness planning and broader efforts to strengthen the health system.

## Figures and Tables

**Table 1 healthcare-13-01333-t001:** Sociodemographic characteristics of participants (*n* = 16).

Characteristic	Category	*n*	Percentage (%)
Gender	Female	10	62.5
	Male	6	37.5
Area of residence	Urban	11	68.7
	Rural	5	31.3
Total participants		16	100

**Table 2 healthcare-13-01333-t002:** Type of healthcare services accessed during the pandemic.

Type of Service	*n*	Percentage%
Public healthcare	8	50.0
Private healthcare	8	50.0

**Table 3 healthcare-13-01333-t003:** Perceived impact of digitalization on service accessibility.

Response	*n*	Percentage%
Access remained the same	11	68.7
Access improved	5	31.3
Access worsened	0	0.00

**Table 4 healthcare-13-01333-t004:** Participant ratings of nursing care quality (1–5 scale).

Rating	*n*
2	4
3	5
4	6
5	1

## Data Availability

The data presented in this study are not publicly available due to ethical and legal considerations related to participant confidentiality. However, de-identified datasets may be made available from the corresponding author and principal investigator upon reasonable request.

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
