# Peer review of "Barriers to Healthcare Access During the Coronavirus Disease 2019 (COVID-19) Pandemic: A Cross-Sectional Study Among Romanian Patients with Chronic Illnesses and Confirmed SARS-CoV-2 Infection"

_healthcare, 2025, doi:10.3390/healthcare13111333_

Round 1
Reviewer 1 Report
Comments and Suggestions for Authors
The study investigates the barriers faced by Romanian chronic illness patients during the COVID-19 pandemic, focusing on healthcare access, digital tools, rural-urban disparities and patient satisfaction with nursing care. Using a cross-sectional study design, the authors identify digital exclusion and rural disparities as key challenges and advocate for inclusive healthcare policies.
Key concerns:
A major limitation is the small sample size (n = 16), reducing the generalizability and statistical power of the conclusions.
Using convenience and snowball sampling could have introduced biases, over-representing digitally literate individuals.
The charts and data presentation are informative, but additional context or comparative data would strengthen the analysis.
Suggestions for improvement:
Even with resource constraints, efforts to include more participants would improve robustness.
Highlight that findings are context-specific and not universally applicable.
Given the significant findings about the digital divide, exploring specific solutions like training programs would add depth.
Discuss more concrete strategies for rural healthcare delivery and digital health integration.
Reviewer 2 Report
Comments and Suggestions for Authors
• The article is interesting and useful at the same time.
• Introduction
• In the introduction I suggest introducing the context in which chronic patients are cared for. Treatment for many of the chronic conditions with an impact on public health is prescribed by family doctors in primary care. To confirm a chronic condition, some diseases require an outpatient consultation (secondary care) where a specialist must confirm the diagnosis and initiate some therapies. The continuation of treatment can be continued by the family doctor in primary care.
This aspect is relevant for interpretation because it is not clear whether the respondents were selected from the hospital (tertiary care), the specialized outpatient clinic (secondary care) or primary care.
• The objectives are well defined but the level of healthcare that was investigated should be mentioned (primary care, secondary care or tertiary care or all).
• Material and methods
• Also, line 82 should mention the level of healthcare evaluated.
• Line 84: Did the patients come from Bucharest or from other regions? Table 1 says that 5 patients were from rural areas. From which county/region of the country? Did the urban patients all come from Bucharest? The area of ​​origin of the patients needs to be defined in order to understand whether the identification barriers are specific to the capital city or can be generalized to the entire country.
• Participants and inclusion criteria: Why was the inclusion criteria “a confirmed history of SARS-CoV-2 infection prior to enrollment” used? How was the sample size calculated so that the results were conclusive? Why only 16 patients?
• Data Collection Instrument: the questionnaire was proposed by the authors. How was this questionnaire validated?
• Results
• Again: was a sample size calculated in order to see the minimum necessary number of responses for a relevant study? Where do the participants come from?
• Table 2: what is the level of healthcare categorized as public or private healthcare?
• Figure 2 duplicates the information in Table 4. Authors should decide which form of presentation of the information they prefer. Duplication of information is not acceptable.
• Line 176 – 179: the statement is evasive for the results chapter. More precisely, how many of the participants reported that delays in medical consul-176 tations and procedures had a negative impact on their health
• The small number of participants would be justified only by a qualitative analysis considering the open-ended questions as stated in lines 100-101. I do not see any analysis of open-ended questions in the results chapter.
Discussion
• I suggest that the first statement briefly summarizes the results.
• In fact, in Romania the services were called remote consultations. Telemedicine was defined later. Recommend replacing the term telemedicine with remote consultation. (line 195)
• Limitations
I suggest including a limitation related to the level of healthcare assessed and the geographical area.
Reviewer 3 Report
Comments and Suggestions for Authors
Though the research area and target subject are of research interest, the primary limitation of this study is the small sample size (N = 16). Drawing robust conclusions from such a limited number of participants poses significant challenges. A sample of this size might be more appropriate for a qualitative study; however, this research does not adopt a qualitative approach, which raises concerns about the validity and generalizability of its findings.
Furthermore, the methods section describes the study as a "cross-sectional observational study," yet there is no indication that observation was utilized as a method of data collection. This discrepancy should be addressed to enhance methodological clarity.
The authors state that both open- and close-ended questions were used during data collection. However, upon reviewing the "SATISFACTION QUESTIONNAIRE" included as a supplementary document, I could not locate any open-ended questions. Including even a few open-ended questions would have enriched the data considerably, as such questions often yield insightful responses that closed-ended questions may miss.
Additionally, the questionnaire begins with demographic questions. Typically, in survey-based research, these questions are placed toward the end to reduce respondent discomfort. Starting with personal questions can discourage participation or lead to incomplete responses. The authors may wish to explain the rationale for this sequencing, as it could have influenced participant engagement and response patterns.
Finally, I recommend expanding the introduction to provide a stronger justification for the study, and enhancing both the discussion and conclusion sections to more thoroughly interpret the findings and align them with results.
Round 2
Reviewer 2 Report
Comments and Suggestions for Authors
No comments
Reviewer 3 Report
Comments and Suggestions for Authors
- The small sample size is the biggest limitation for this study and paper. I recommend to cite and/or discuss similar studies or publications that have employed small samples like in this to justify the methodological approach.
- Offering public health policy recommendations based on such a limited sample is challenging to substantiate without appropriate contextual references.
Altough overall writing is good, there are statements requiring attention or rewording. For example, the statement (lines: 127-128) "The questionnaire was administered directly by the principal investigator, ...". This sentences can be presented as "The questionnaire was administered by the principal investigator after obtaining written or verbal informed consent from each participant." There are places throughout the paper requiring similar improvement.
